

# Genome-wide identification and expression profile analysis of *CCH* gene family in *Populus*

Zhiru Xu[1,2], Liying Gao[1], Mengquan Tang[1], Chunpu Qu[2,3], Jiahuan Huang[1], Qi Wang[2,3], Chuanping Yang[2,3], Guanjun Liu[2,3] and Chengjun Yang[3]

[1] College of Life Science, Northeast Forestry University, HarBin, China
[2] State Key Laboratory of Tree Genetics and Breeding, Northeast Forestry University, HarBin, China
[3] School of Forestry, Northeast Forestry University, HarBin, China

## ABSTRACT

Copper plays key roles in plant physiological activities. To maintain copper cellular homeostasis, copper chaperones have important functions in binding and transporting copper to target proteins. Detailed characterization and function analysis of a copper chaperone, CCH, is presently limited to *Arabidopsis*. This study reports the identification of 21 genes encoding putative CCH proteins in *Populus trichocarpa*. Besides sharing the conserved metal-binding motif MXCXXC and forming a βαββαβ secondary structure at the N-terminal, all the PtCCHs possessed the plant-exclusive extended C-terminal. Based on their gene structure, conserved motifs, and phylogenetic analysis, the PtCCHs were divided into three subgroups. Our analysis indicated that whole-genome duplication and tandem duplication events likely contributed to expansion of the *CCH* gene family in *Populus*. Tissue-specific data from PlantGenIE revealed that *PtCCH* genes had broad expression patterns in different tissues. Quantitative real-time RT-PCR (qRT-PCR) analysis revealed that *PnCCH* genes of *P. simonii* × *P. nigra* also had different tissue-specific expression traits, as well as different inducible-expression patterns in response to copper stresses (excessive and deficiency). In summary, our study of *CCH* genes in the *Populus* genome provides a comprehensive analysis of this gene family, and lays an important foundation for further investigation of their roles in copper homeostasis of poplar.

## INTRODUCTION

Copper (Cu) is an important transition metal in plants, and is involved in many physiological processes. As a cofactor, copper is required for plastocyanin, cytochrome c oxidase (or complex IV), copper/zinc superoxide dismutase (Cu/Zn SOD), the ethylene receptor, diamine oxidase, laccase, multicopper ferroxidase, and polyphenol oxidase (*Pilon et al., 2006*; *Burkhead et al., 2009*; *Yamasaki et al., 2009*). In the cell environment, Cu can exist in two states, Cu(II) and Cu(I). Free Cu ions in cells can generate reactive oxygen species (ROS) such as superoxide, hydrogen peroxide ($H_2O_2$), and the hydroxyl radical, which then damage biomacromolecules such as proteins, lipids, and DNA

Corresponding authors
Guanjun Liu,
Liuguanjun2013@nefu.edu.cn
Chengjun Yang, nxyycj@sina.cn

(*Harrison, Jones & Dameron, 1999*; *Brewer, 2010*). To protect cells against these types of free radical-mediated damage induced by excess copper, organisms have developed the capacity to scavenge or invalidate ROS through the combined action of highly specialized antioxidant proteins in the evolutionary process (*Lin & Culotta, 1995*).

To appropriately regulate copper homeostasis in response to environmental copper levels, plants have developed a complex network of metal trafficking pathways to ensure proper delivery of this essential element to target metalloproteins (*Yruela, 2009*). This network involves the acquisition and distribution of copper. Recent reports have indicated that many key components are involved in this process in plants, including copper transporters, copper chaperones, and copper proteins (*Yruela, 2005*; *Burkhead et al., 2009*). The COPT/Ctr family members are membrane proteins that act as copper transporters. The first cloned copper transporter was ScCTR1 from *Saccharomyces cerevisiae*, after which many homologous proteins have been identified in plants (*Dancis et al., 1994*; *Sancenón et al., 2003*; *Martins et al., 2012*). *Arabidopsis thaliana* has six AtCOPT proteins that play important roles in copper acquisition in response to deficiency or excess copper levels, copper export from the vacuole, and distribution of copper from root to aerial tissues (*Sancenón et al., 2004*; *Klaumann et al., 2011*; *Perea-García et al., 2013*). The COPT/Ctr family in rice contains seven members, among which OsCOPT2, OsCOPT3, or OsCOPT4 may cooperate with OsCOPT6 for high-affinity copper uptake, while OsCOPT7 may act alone to transport copper (*Yuan et al., 2011*).

After its transfer into the cytoplasm, copper is delivered to specific organelles and/or to specific copper proteins to protect cells from copper toxicity. The copper chaperones function as Cu(I) ion shuttles to transport copper to their target proteins in various intracellular compartments (*Harrison, Jones & Dameron, 1999*; *Markossian & Kurganov, 2003*). At least three types of copper chaperones have been found in Arabidopsis. The copper chaperone for superoxide dismutase delivers copper to Cu/Zn SODs, including CSD1 and CSD3 in the cytosol, and CSD2 in the plastid (*Casareno, Waggoner & Gitlin, 1998*; *Chu et al., 2005*). The AtCOX17 chaperone supplies copper to mitochondria to assemble a functional cytochrome oxidase complex (*Balandin & Castresana, 2002*). Other mitochondrial copper chaperones, COX19 and COX11, are also required for cytochrome c oxidase (COX) activation in plants (*Carr & Winge, 2003*; *Attallah et al., 2007*; *Radin et al., 2015*). The third types of copper chaperones are the homologs of yeast Atx1, AtCCH and AtATX1 (*Himelblau et al., 1998*; *Puig et al., 2007a*). AtATX1 interacts with responsive-to-antagonist 1 (RAN1; also known as HMA7) and HMA5; RAN1 is required for ethylene signaling, and HMA5 contributes to copper efflux in Arabidopsis (*Hirayama et al., 1999*; *Andrés-Colás et al., 2006*). Therefore, AtATX1 is likely related to copper homeostasis and ethylene signal transduction.

AtCCH was the first copper chaperone identified in Arabidopsis. It shares 36% similarity with Atx1 of *S. cerevisiae* and 74% similarity with AtATX1 (*Andrés-Colás et al., 2006*). Besides the conserved βαββαβ-fold structure and a MXCXXC Cu$^+$-binding motif in its N-domain, AtCCH has the plant-specific C-terminal extended conformation with special amino acid composition. The N-domain of CCH could rescue the high-affinity iron uptake function of yeast *atx1Δ* cells, as well as the oxidation resistance function of

yeast *sod1Δ* cells (*Himelblau et al., 1998*; *Puig et al., 2007a*). These functions are copper dependent in yeast mutant cells. Copper deficiency, senescence, and oxidative stress upregulated the expression of *AtCCH*, and *CCH* mRNA decreased rapidly under excess copper conditions, but the CCH protein remained stable (*Himelblau et al., 1998*; *Puig et al., 2007a*). Moreover, CCH was located mainly around the vascular bundles of senescing leaves and petioles, and also accumulated in stem sieve elements (*Mira, Martínez-García & Peñarrubia, 2001a*). These findings indicate that CCH might be transported through plasmodesmata to nonnucleated cells by means of its C-terminal domain. In this way, CCH could deliver copper during senescence-associated nutrient mobilization through the symplastic pathway. Otherwise, the C-domain of CCH adopted an extended conformation in solution and formed well-ordered amyloid-like fibrils (*Mira et al., 2004*). Further, the C-domain, not the N-domain, altered the SDS-PAGE mobility of the whole CCH protein (*Mira et al., 2001b*; *Puig et al., 2007a*). After deleting the C-terminus, CCH could interact with the metal binding domain of the P-type ATPases, RAN1 and HMA5 (*Puig et al., 2007b*; *Shin, Lo & Yeh, 2012*). These results suggest that the CCH C-terminal extension might possess special functions in copper homeostasis of plant. Although the CCH protein has been investigated to some extent in *Arabidopsis*, few studies of this protein have been reported in woody plants.

Large and perennial forest trees have massive root systems, extensive secondary growth, a coordinated signaling system, and powerful adaptability against physical and biotic stresses. All of these characteristics distinguish trees from annual herbaceous plants. The genome sequences of *Populus trichocarpa* (Torr. & Gray) was published in 2006, and from then on, *Populus* has been selected as a model tree system for genetic research (*Taylor, 2002*; *Tuskan et al., 2006*; *Zhang et al., 2015*). *Populus* species have been used as heavy metal phytoremediation plants as they are adaptable to grow on contaminated areas and able to accumulate metals than short-lived plants (*Di Loardo et al., 2011*). *Populus simonii* × *Populus nigra* is a hybrid poplar that is widely planted in northern China, and is widely used for afforestation and commercial forestry (*Chen et al., 2012*). To analyze copper homeostasis in woody plants, we investigated the copper chaperone genes in *Populus*.

In this study, to identify the *CCH* gene family members of *P. trichocarpa*, we conducted a genome-wide analysis using bioinformatics method. 21 *PtCCH* genes were obtained, and the multiple alignments of the deduced amino acid sequences indicated that all of them had the metal-binding motif and the typical secondary structure at the N-terminal. PtCCHs belonged to the same subgroup often had the similar gene structures and motifs. The results of quantitative real-time RT-PCR (qRT-PCR) showed that the expression of the *CCH* genes in *P. simonii* × *P. nigra* could be induced or suppressed by deficiency or excessive copper conditions. We also found that the *CCH* genes of *Populus* had tissue-specific expression profiles. Our results may provide insights into the roles of CCH proteins of *Populus* in copper homeostasis.

## MATERIALS AND METHODS

### Identification of CCH family genes in *P. trichocarpa*

To identify CCH proteins in *P. trichocarpa,* the protein database of *Populus* was downloaded from https://phytozome.jgi.doe.gov/pz/portal.html (*Goodstein et al., 2012*), and Hidden Markov Model (HMM) profile file (HMA.hmm) of the Heavy-metal-associated domain (HMA) (PF00403) was downloaded from the Pfam database (http://pfam.xfam.org/). The HMA.hmm file was exploited as a query to identify PtCCHs in the poplar protein database using the hmmer search command of the HMMER (v 3.0) software. On the other hand, *A. thaliana* AtCCH (AT3G56240) was used as the query sequence in a BlastP search against the *P. trichocarpa* genome database online. The relative parameters were as follows: target type = Proteome, Program = BLASTP-protein query to protein db, Expect (E) threshold = −5; the other parameters were the default. The sequences obtained by the above two methods were used for the subsequent selection, and the screening criteria was whether the protein had one conserved MXCXXC (M is methionine, X is any amino acid, C is cysteine) site (*Arnold et al., 2006*; *Kiefer et al., 2009*; *Guex, Peitsch & Schwede, 2009*; *Biasini et al., 2014*) and formed one βαββαβ structure in the N-terminal (SWISS-MODEL (http://swissmodel.expasy.org/) was used to construct the secondary structure), as well as the extended segment in the C-terminal. As AtCCH is a small cytoplasmic protein and has no identified functional domains in its C-terminal, we deleted the proteins with long C-terminal sequences which might influence the function of the metal binding domain in the N-terminal when formed the tertiary structures, as well as the proteins had the domains in their C-terminal if the functions of the domains have been identified. After eliminating those did not meet these criteria, the 21 remainders were identified as PtCCHs and the amino acid sequences were analyzed for the characteristics using online ExPASy programs (http://www.expasy.org/), such as molecular weight, isoelectric point, number of amino acids, aliphatic index, and grand average of hydropathicity (GRAVY) score (*Brunner, Busov & Strauss, 2004*; *Gasteiger et al., 2005*).

### Exon/intron structure and conserved motifs analysis

The distribution patterns of exons and introns in the *PtCCH* genes were predicted using the Gene Structure Display Server (GSDS2.0, http://gsds.cbi.pku.edu.cn) (*Hu et al., 2015*), and the online MEME tool (Multiple Em for Motif Elicitation, version 4.11.3, http://meme-suite.org/tools/meme) (*Bailey et al., 2009*) was used to identify conserved motifs in the PtCCHs. The setting parameters were as follows: maximum number of different motifs to find = 10, minimum width = 7, maximum width = 50.

### Multiple sequence alignment and phylogenetic analysis

The PtCCH amino acid sequences were aligned using the Clustal X program (*Thompson et al., 1997*) and the conserved sites of MXCXXC were checked (*Himelblau et al., 1998*; *Puig et al., 2007a*). MEGA5.0 software (*Tamura et al., 2011*) was used to construct phylogenetic trees by the Neighbor-Joining (NJ) method (No of bootstrap replications were 1,000) with the full predicted CCH amino acid sequences of *P. trichocarpa*, *A. thaliana*, and *O. sativa*. *AtCCHs* and *OsCCHs* were identified using the same method and the same criteria as

in *P. trichocarpa*, and the sequences of AtCCHs and OsCCHs were retrieved from their genome databases (*Goodstein et al., 2012*).

## Mapping *PtCCH* genes on *Populus* chromosomes and subcellular locations of PtCCHs

The location information of *PtCCH* genes were retrieved from Phytozome and PopGenIE (http://popgenie.org/chromosome-diagram), and then the chromosomal location maps were constructed (*Sjödin et al., 2009*). The positions of whole-genome duplication blocks of different chromosomes were defined according to the study of *Tuskan et al. (2006)*. The subcellular locations of PtCCH proteins were predicted using the online WoLF PSORT tool (http://www.genscript.com/psort/wolf_psort.html) (*Horton et al., 2007*), and the option of organism type was plant.

## Identification of tissue-specific expression patterns of *PtCCH* genes

The tissue-specific expression data of *PtCCH* genes in mature leaves, young leaves, roots, nodes, and internodes were derived from PopGenIE (http://popgenie.org) (*Sundell et al., 2015*), and used to generate visual images.

## Plant materials, growth conditions, and copper stress treatments

*P. simonii* × *P. nigra* were grown at Northeast Forestry University Forest Farm, Harbin, China. Cuttings of *P. simonii* × *P. nigra* were cultivated in pots containing sterile mixture of turfy soil, vermiculite, and perlite (3:2:1) in the greenhouse under long-day conditions (16 h light/8 h dark) at 25 °C, and supplied with water every 3 days. The seedlings were cut when they were 10 cm tall and then rooted in water that was aerated constantly by aquarium pumps. After rooting, the plantlets were transferred into sterile vermiculite and supplied every 2 days with half-strength modified Hoagland (*Assunção et al., 2003*) nutrient solution composition of: 3 mM $KNO_3$, 2 mM $Ca(NO_3)_2 \cdot 4H_2O$, 1 mM $NH_4 H_2PO_4$, 0.5 mM $MgSO_4 \cdot 7H_2O$, 25 µM KCl, 12.5 µM $H_3BO_3$, 1 µM $MnSO_4 \cdot H_2O$, 1 µM $ZnSO_4 \cdot H_2O$, 0.247 µM $Na_2MoO_4$, 10 µM NaFeDTPA (Fe), and 0.5 µM $CuSO_4 \cdot 5H_2O$; the pH was fixed at 5 with KOH. After 2 weeks, seedlings were treated as follows: To test the expression profiles response to copper stress, seedlings were transferred into half-strength modified Hoagland nutrient solution that contained 0, 0.5 (control), or 10 µM Cu applied as $CuSO_4 \cdot 5H_2O$. Plants were treated at different times in the day, and were maintained in these conditions for 3, 12, or 72 h, then roots, stems, young leaves (LPI 0–2), and mature leaves (LPI 7–9) (*Larson & Isebrands, 1971*) were harvested at the same time, 10 o'clock. To determine the tissue-specific expression patterns, the plant materials were collected directly without treatment at 10 o'clock as described above. The samples were frozen in liquid nitrogen and stored at −80 °C for further analysis. To obtain reproducible results, three biological replicates of each stress treatment were conducted, and each experiment was repeated three times with independent samples.

## RNA extraction and qRT-PCR analysis

Total RNA was extracted from leaves, stems, and roots using pBIOZOL Plant Total RNA Extraction Reagent (BioFlux, Hangzhou, China) according to the protocol. After removing

genomic DNA, first-strand of cDNA was obtained using a PrimeScript$^{TM}$ RT reagent Kit (Takara Bio, Dalian, China). qRT-PCR was performed using UltraSYBR Mixture (Low ROX) (CWBIO, Beijing, China) with three replicates on an ABI 7500 Real Time PCR System (Applied Biosystems, Foster City, CA, USA). Amplifications were conducted in 20 μL reaction mixtures containing 10.0 μL 2 × UltraSYBR Mixture (with ROX), 4.0 ng cDNA template, and 0.2 μM of forward and reverse primers. The primers used for qRT-PCR were designed with Primer Premier 5.0 (Premier Biosoft, Palo Alto, CA, USA) according to *PtCCH* gene sequences (Table S1), and the annealing temperature was between 46 °C and 60 °C. The *PtUBQ7* (XM_002306689.2) and *PtCDC2* (XM_002305968.2) genes of *P. trichocarpa* were used as reference genes (*Pettengill, Parmentier-Line & Coleman, 2012*; *Pettengill, Pettengill & Coleman, 2013*), and the expression levels of them were stable, so we selected *PtUBQ7* (XM_002306689.2) gene as reference gene to analyze the expression relative changes of *PtCCH* genes. The amplification conditions were as follows: denaturation at 95 °C for 10 min, 45 cycles of denaturation at 95 °C for 15 s, annealing between 60 °C and 68 °C for 60 s.

The $2^{-\Delta\Delta CT}$ method was used to analyze the expression levels of *PnCCHs* in different tissues under control copper condition (0.5 μM $CuSO_4$) or copper stress conditions (0 μM $CuSO_4$ and 10 μM $CuSO_4$) (*Livak & Schmittgen, 2001*; *Schmittgen & Livak, 2008*). The statistical significant differences of the tissue-specific expression profiles of *PnCCHs* were analyzed using duncan's test ($P < 0.05$). The values of $\log_2$ (sample/control) copper stress conditions and different treated times were calculated as the relative expression levels of every *PnCCH* genes, and the different color of the cells in the heatmaps indicated the expression level of the treated samples went up or down compared with their controls. The statistical significant differences of the expression levels were analyzed using $t$ test ($P < 0.05$). All the values under copper stress conditions were compared with the corresponding values under control copper condition. Results are the mean ± SD of three replicates.

## RESULTS

### Identification and sequence conservation of *CCH* genes in *P. trichocarpa*

A total of 21 *PtCCH* genes (named *PtCCH1–21*) were identified in the *P. trichocarpa* genome according to their encoded proteins that contained the conserved domains and typical secondary structures like the AtCCH of *A. thaliana*. The 21 *PtCCH* genes were analyzed and the parameters, including locus name, chromosome location, protein length, molecular weight, isoelectric point, aliphatic index, and grand average of hydropathicity (GRAVY) score, are listed in Table 1. The putative PtCCH protein sequences varied from 115 to 179 amino acids (aa) in length, the isoelectric points ranged from 4.96 to 9.64, and all of them were hydrophilic because of the GRAVY values were negative. The 21 PtCCH amino acid sequences shared 23–100% identities (Table S2), and the proteins were predicted to be localized in cytosol, chloroplast, or mitochondria.

The multiple sequence alignment was conducted to examine the conserved domains of the PtCCHs, and the result showed that all the PtCCH proteins shared the consensus

Xu et al. (2017), *PeerJ*, DOI 10.7717/peerj.3962

**Table 1** Parameters for the 21 identified *CCH* genes and deduced polypeptides present in the *P. trichocarpa* genome.

| Gene name | Locus name Phytozome v3.0 | Amino acid no. | Molecular weight (Da) | Isoelectric points | Aliphatic index | GRAVY | Chromosome location | Cellular localization |
|---|---|---|---|---|---|---|---|---|
| *PtCCH1* | Potri.001G452400.1 | 179 | 20,864.8 | 9.01 | 66.93 | −0.55 | ChrI:48764846..48766902 | Cytosol |
| *PtCCH2* | Potri.001G468500.1 | 115 | 12,727.9 | 8.85 | 86.43 | −0.397 | ChrI:50097242..50098160 | Cytosol |
| *PtCCH3* | Potri.002G092200.1 | 150 | 16,758.5 | 9.64 | 72 | −0.48 | ChrII:6570956..6572016 | Chloroplast |
| *PtCCH4* | Potri.004G056800.1 | 146 | 17,269.3 | 4.96 | 57.47 | −0.864 | ChrIV:4523350..4524675 | Mitochondria |
| *PtCCH5* | Potri.005G003700.1 | 150 | 16,782.6 | 9.33 | 82.4 | −0.254 | ChrV:226035..226652 | Chloroplast |
| *PtCCH6* | Potri.005G079800.1 | 153 | 16,953.4 | 9.48 | 72.48 | −0.59 | ChrV:5867660..5868994 | Chloroplast |
| *PtCCH7* | Potri.005G110400.1 | 157 | 17,554.1 | 6.31 | 75.61 | −0.455 | ChrV:8497273..8498140 | Chloroplast |
| *PtCCH8* | Potri.005G167000.1 | 150 | 16,647.3 | 9.44 | 70.73 | −0.447 | ChrV:17596661..17597639 | Chloroplast |
| *PtCCH9* | Potri.005G169700.1 | 150 | 16,647.3 | 9.44 | 70.73 | −0.447 | ChrV:18274591..18275572 | Chloroplast |
| *PtCCH10* | Potri.006G006100.4 | 163 | 18,702.2 | 5.28 | 69.26 | −0.817 | ChrVI:415310..419144 | Chloroplast |
| *PtCCH11* | Potri.006G006100.5 | 136 | 15,360.5 | 6.27 | 60.88 | −1.307 | ChrVI:416730..419039 | Chloroplast |
| *PtCCH12* | Potri.006G024800.2 | 150 | 17,735 | 6.51 | 62.93 | −0.597 | ChrVI:1722458..1723578 | Cytosol |
| *PtCCH13* | Potri.007G087300.1 | 151 | 16,667.1 | 9.47 | 70.93 | −0.514 | ChrVII:11343882..11345554 | Chloroplast |
| *PtCCH14* | Potri.010G015300.1 | 137 | 15,673.2 | 9.16 | 66.86 | −0.678 | ChrX:1921667..1922272 | Cytosol |
| *PtCCH15* | Potri.010G114600.1 | 150 | 16,820.3 | 9.63 | 66.2 | −0.605 | ChrX:13289528..13291583 | Chloroplast |
| *PtCCH16* | Potri.010G114600.2 | 130 | 14,482.6 | 9.41 | 60.62 | −0.498 | ChrX:13289596..13291583 | Chloroplast |
| *PtCCH17* | Potri.011G065600.1 | 147 | 17,353.5 | 7.05 | 49.12 | −0.94 | ChrXI:6016497..6017737 | Mitochondria |
| *PtCCH18* | Potri.011G065600.2 | 142 | 16,777.9 | 7.74 | 48.1 | −0.984 | ChrXI:6016577..6017733 | Mitochondria |
| *PtCCH19* | Potri.011G149500.1 | 178 | 20,665.6 | 9.03 | 72.81 | −0.477 | ChrXI:16795622..16798712 | Cytosol |
| *PtCCH20* | Potri.017G123400.1 | 146 | 16,363.8 | 9.49 | 77.26 | −0.311 | ChrXVII:13590414..13592054 | Cytosol |
| *PtCCH21* | Potri.019G106500.1 | 140 | 16,071.8 | 7.23 | 54.93 | −0.924 | ChrXIX:13509997..13510615 | Chloroplast |

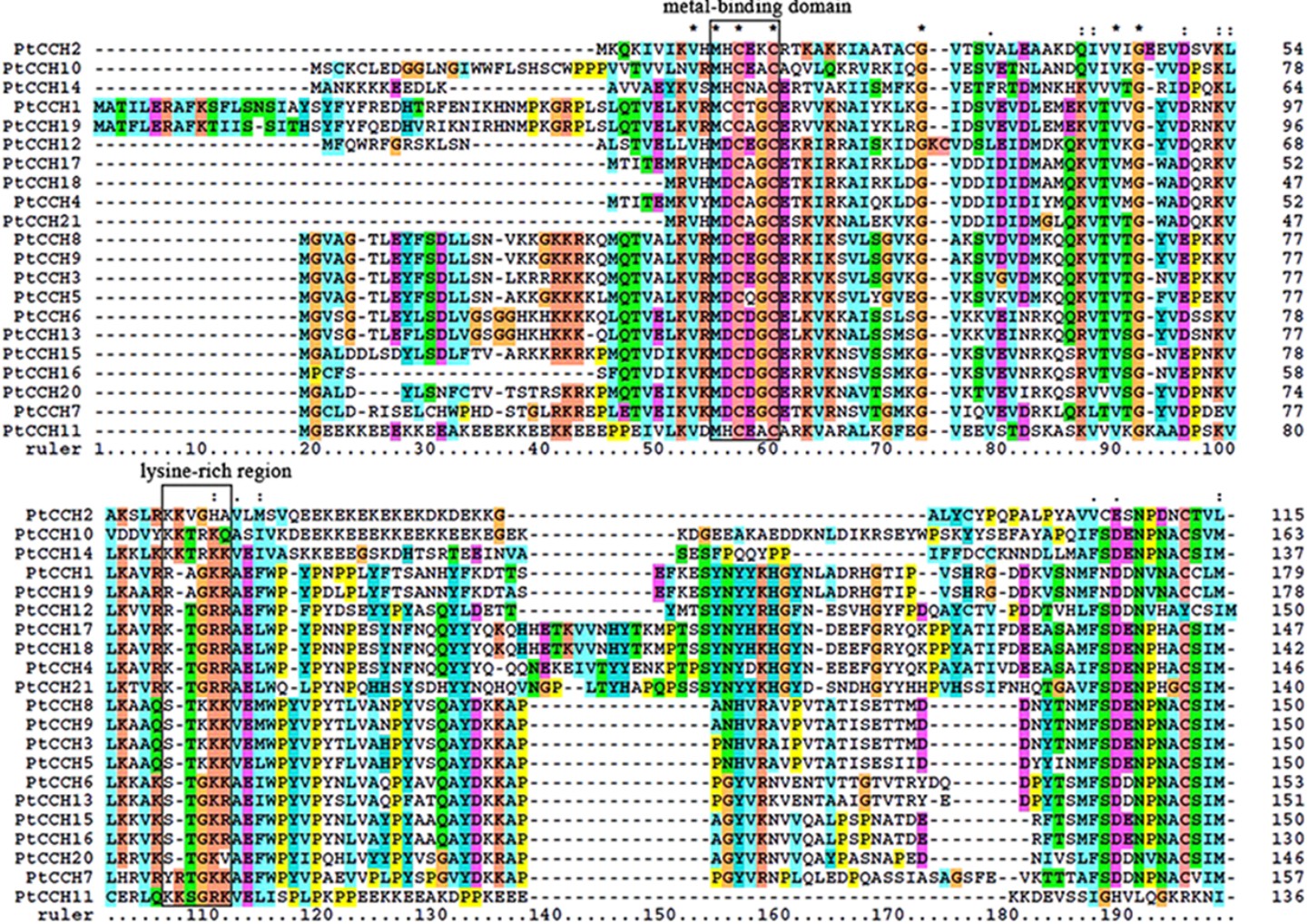

**Figure 1** **Multiple alignment of the deduced amino acid sequences of the 21 PtCCHs.** The boxes indicate the conserved MXCXXC (M is methionine, X is any amino acid, and C is cysteine) metal-binding site and lysine-rich region. Asterisks indicate the identical amino acids, and dashes indicate gaps introduced to optimize the alignment. The alignment was performed using the Clustal X software.

metal-binding motif MXCXXC in the N-terminal, and some PtCCHs had a lysine-rich (Lys, K) region (Fig. 1) (*Lin & Culotta, 1995*; *Agrawal et al., 2002*; *Lee et al., 2005*). These two structure characteristics are involved in the function of copper chaperones (*Rosenzweig et al., 1999*; *Mira et al., 2001b*). The secondary structure prediction indicated that all the PtCCHs possessed, at their N-terminal, the conserved βαββαβ-fold structure that is present in the ATX1-like family of metallochaperones (Fig. S1).

## Chromosomal location and duplication of *PtCCHs* in *P. trichocarpa*

To study the gene distribution and duplication in the *P. trichocarpa* genome, we mapped all of the identified *PtCCH* genes to their corresponding chromosomes. The 21 *PtCCH* genes were found to be physically located on 10 of the 19 *Populus* chromosomes (none on chromosomes III, VIII, IX, XII, XIII, XIV, XV, XVI, and XVIII) (Fig. 2). The distribution

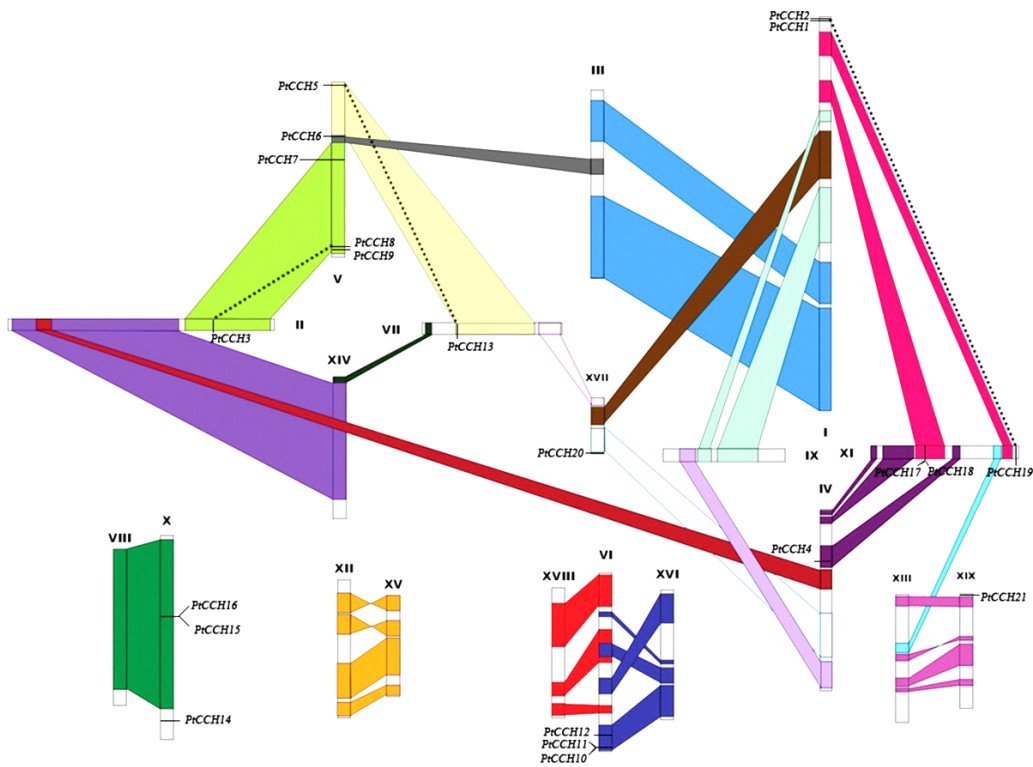

**Figure 2** **Chromosomal locations of the 21 *PtCCH* genes in the *P. trichocarpa* genome.** The schematic diagram of chromosome recombination was constructed using the data from the whole-genome duplication study of *Tuskan et al. (2006)*. Common colors refer to homologous genome blocks which are presumed to have arisen from the salicoid-specific genome duplication shared by two chromosomes. Paralogous *CCH* genes are indicated by dashed lines.

of the *PtCCH* genes on each chromosome was obviously heterogeneous: ChrV harbored the highest number (5) of *PtCCH* genes; ChrI harbored two *PtCCH* genes (*PtCCH1* and *PtCCH2*); ChrVI, ChrX, and ChrXI harbored three *PtCCH* genes each; while only a single *PtCCH* gene was found on ChrII, ChrIV, ChrVII, ChrXVII, and ChrXIX.

According to the study of *Tuskan et al. (2006)*, a whole-genome duplication (WGD) event occurred 60–65 million years ago in the *Salicaceae* family (salicoid duplication), and this created the identified paralogous segments in the *Populus* genome. Except for six genes (*PtCCH1*, *2*, *14*, *19*, *20*, *21*), fifteen *PtCCH* genes were located in duplicate blocks (Fig. 2). Five pairs of genes (*PtCCH1/2*, *PtCCH8/9*, *PtCH10/11*, *PtCCH15/16*, and *PtCCH17/18*) were arranged in tandem repeats on ChrI, ChrV, ChrVI, ChrX, and ChrXI, respectively.

## Phylogenetic analysis of the *PtCCH* gene family

We constructed a phylogenetic tree to analyze the evolutionary relationships of the *PtCCH* genes and to classify the *PtCCHs* in *P. trichocarpa*. Twenty-one full-length PtCCH sequences were used to generate an unrooted phylogenetic tree (Fig. 3A). The 21 sequences clustered into three distinct subgroups: SI, SII, and SIII. The SI subgroup contained PtCCH3, 5, 6,
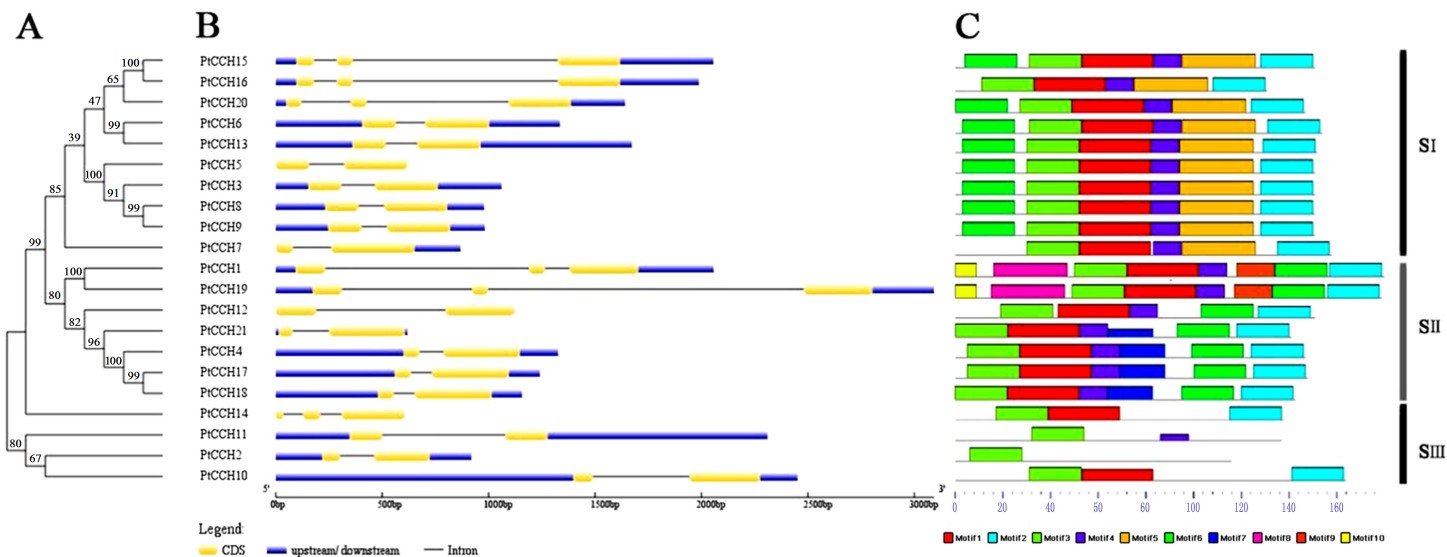

**Figure 3** **Phylogenetic tree and structure analysis of the 21 *PtCCH* genes.** (A), Phylogenetic tree based on the deduced full-length amino acid sequences of the PtCCHs constructed by the Neighbor-Joining (NJ) method. (B), Structure of the corresponding *PtCCH* genes. Yellow indicates CDSs, blue indicates upstream/downstream sequences, and black line indicates introns. (C), Predicted motifs in the PtCCH amino acid sequences predicted using the online MEME tool. The sequences of the motifs are listed in Table S4.

7, 8, 9, 13, 15, 16, and 20; the SII subgroup contained PtCCH1, 4, 12, 17, 18, 19, and 21; and the SIII subgroup contained the remaining four PtCCHs.

To analyze the relationships of the PtCCHs to CCHs in other species, we constructed a phylogenetic tree using full-length amino acid sequences of the 21 *P. trichocarpa* CCHs, 13 *A. thaliana* CCHs (AtCCH6 was the AtCCH which has been functionally characterized and used as the query sequence to screen the *CCH* genes), and 12 *Oryza sativa* CCHs (Table S3). All these CCH sequences clustered into three groups: SI, SII, and SIII. The SI and SII groups contained 22 and 16 members, respectively, and the SIII group had eight members (Fig. 4). The SI group had four subgroups, SI–1, SI–2, SI–3, and SI–4; the SII group had two subgroups, SII–1 and SII–2; and the SIII group had three subgroups, SIII–1, SIII–2, and SIII–3. Among the three species, CCHs that had the same and similar number of motifs clustered into the same clade. The phylogenetic analysis showed, noticeably, subgroup SI–3 contained only PtCCHs, and subgroup SIII–1 contained AtCCH and OsCCH but no PtCCH. Otherwise, the CCH proteins encoded by paralogous pairs of *PtCCH* genes clustered together in the same subgroups; for instance, PtCCH3 and PtCCH8/9 were assigned to SI-3, and PtCCH1 and PtCCH19 were assigned to SII–1. Also, the products of the tandem duplication pairs, PtCCH8/9 and PtCCH17/18, were assigned to SI–3 and SII–2, respectively.

**Analyses of gene structure and conserved motif distribution**
To gain further insight into the possible relationships among different gene family members, the exon/intron organization of the *PtCCH* genes was compared in the context of the phylogenetic tree (Fig. 3B). The relative lengths of the introns and exons were provided

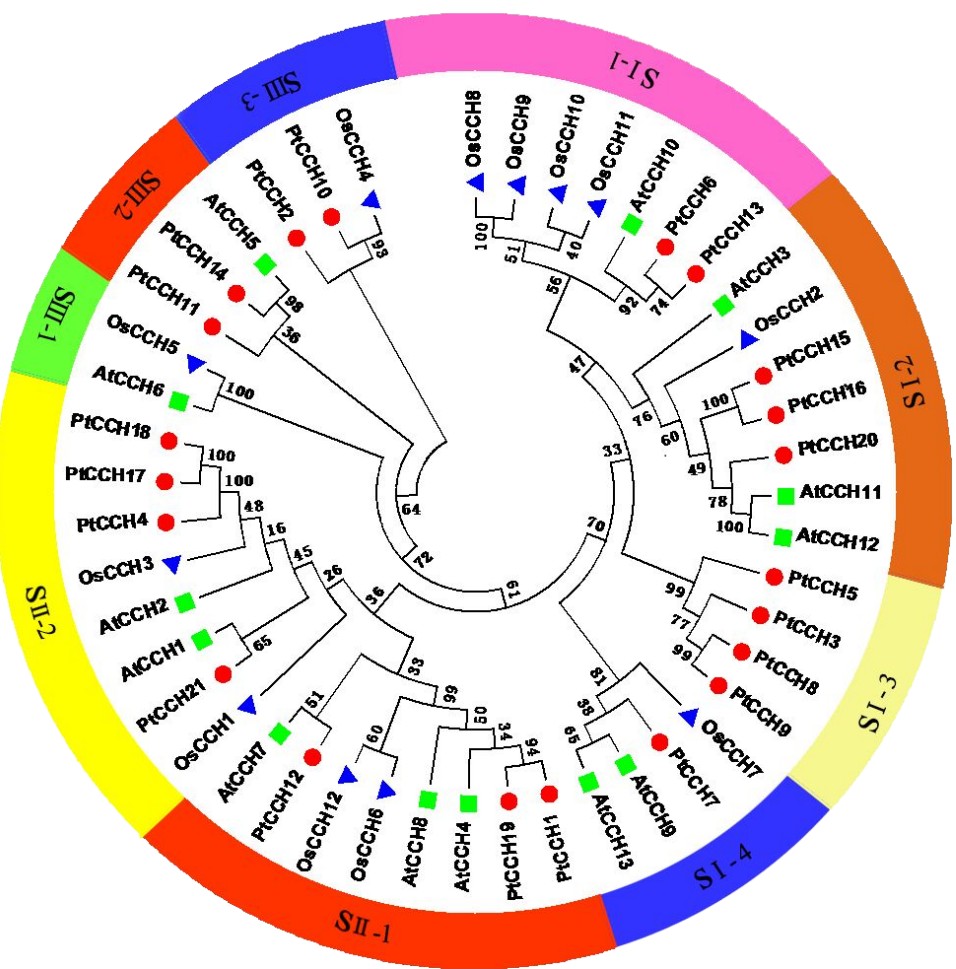

**Figure 4  Phylogenetic analysis of CCH proteins from *P. trichocarpa*, *A. thaliana*, and *O. sativa*.** The deduced full length amino acid sequences were used to generate the phylogenetic tree by the Neighbor-Joining (NJ) method. Red dots indicate *P. trichocarpa* CCHs, green squares indicate *A. thaliana* CCHs, and blue triangles indicate *O. sativa* CCHs.

detailed, and revealed a highly-conserved distribution of intron regions (from one to two in numbers) among the *PtCCH* genes, which is similar to the distribution of intron regions among *Arabidopsis CCH* genes (one to two introns) and *O. sativa CCH* genes (null to three introns). Fifteen of the 21 *PtCCH* genes had only one intron, and the other six had two introns. Notably, closely related genes generally had similar exon/intron structures, although the lengths of the introns and exons varied. There were some exceptions when compared with the other members of the same subgroup; for example, *PtCCH15*, *PtCCH16*, and *PtCCH20* in SI, *PtCCH1* and *PtCCH19* in SII, and *PtCCH14* in SIII. These results imply a tight relationship between the phylogeny and gene structure of the *PtCCHs*, and the regularity of the gene structure might be related to the evolutionary trends and the conservation of the gene family members.
Ten motifs (named motif 1–10) were predicted in the PtCCH sequences using MEME software (Table S4). As shown in Fig. 3C, the PtCCH proteins in the same subgroup generally contained similar motifs. Motif 3, which was present in all the PtCCHs, contains the MXCXXC site which is essential for copper binding. Most members of the PtCCH family shared three motifs, motif 3, motif 1, and motif 2, which were linked in the same order. A few proteins, such as PtCCH2 and PtCCH11, showed quite different protein structures compared with the others. In SI, except PtCCH7 and PtCCH16 without motif 6, all the other PtCCHs showed the same motif distribution and arrangement. Motif 5 appeared only in the SI subgroup. In SII, except PtCCH12 without motif 7, the other motifs in PtCCH4, 12, 17, 18, and 21 were the same. Interestingly, motif 10 was selectively distributed among a specific pair, PtCCH1 and PtCCH19, which had the same motif types and arrangements. The PtCCHs in subgroup SIII had few motifs (one to three) compared with the PtCCHs in the SI and SII subgroups. Overall, the PtCCHs in specific subgroups appear to have evolved conserved and diverged motifs during evolution.

## Tissue-specific expression profiles of *CCH* genes in the *Populus* genome

We used the tissue-specific expression data of *PtCCH* genes in mature leaves, young leaves, roots, nodes, and internodes from the Plant Genome Integrative Explorer (http://PlantGenIE.org) (*Sundell et al., 2015*) to generate visual images (Figs. 5A–5R). The expression data showed that *PtCCH2*, *PtCCH10*, *PtCCH11*, *PtCCH12*, *PtCCH14*, *PtCCH15*, and *PtCCH16* were highly expressed in mature leaves; *PtCCH3*, *PtCCH6*, *PtCCH8*, *PtCCH17*, *PtCCH18*, and *PtCCH20* were highly expressed in roots; and *PtCCH1* and *PtCCH19*, which were clustered into the same clade, were highly expressed in roots/internodes/nodes. The expression levels of *PtCCH4*, *PtCCH5*, *PtCCH7*, *PtCCH13* were higher in mature leaves/roots, young/mature leaves, young leaves, internodes, respectively, and *PtCCH21* was highly expressed in roots, and young and mature leaves. Expression data for *PtCCH9*, which has the same nucleotide sequences as *PtCCH8*, was not available in http://PlantGenIE.org.

Tissue-specific expression patterns of *PnCCHs* in *P. simonii* × *P. nigra* roots, stems, and mature and young leaves were measured and analyzed by qRT-PCR. The primers were designed based on the nucleic acid sequences of the *PtCCHs* (Table S1). *PtCCH8* and *PtCCH9* have identical sequences and may be tandem repeats (Fig. 2). *PtCCH15* and *PtCCH16* are the different transcripts that were described in the gene information of *P. trichocarpa* genome database and their putative proteins have highly similar amino acid sequences, as well as *PtCCH17* and *PtCCH18*; therefore, we did not design primers for *PtCCH16* and *PtCCH18*. We could not design primers for *PtCCH21* because its nucleotide sequences shared high identity with other genes in the *Populus* database (*Tuskan et al., 2006*). Beside this, we did not obtain the amplification results of *PnCCH13* and *PnCCH14*. So, the primers for *PtCCH9*, *13*, *14*, *16*, *18*, and *21* genes are not included in Table S1. The results showed that *PnCCH3*, *PnCCH4*, *PnCCH6*, *PnCCH7*, *PnCCH8*, *PnCCH17*, and *PnCCH20* were most highly expressed in roots, and *PnCCH1*, *PnCCH5*, and *PnCCH15* were most highly expressed in mature leaves. *PnCCH10* and *PnCCH11*, *PnCCH12* and

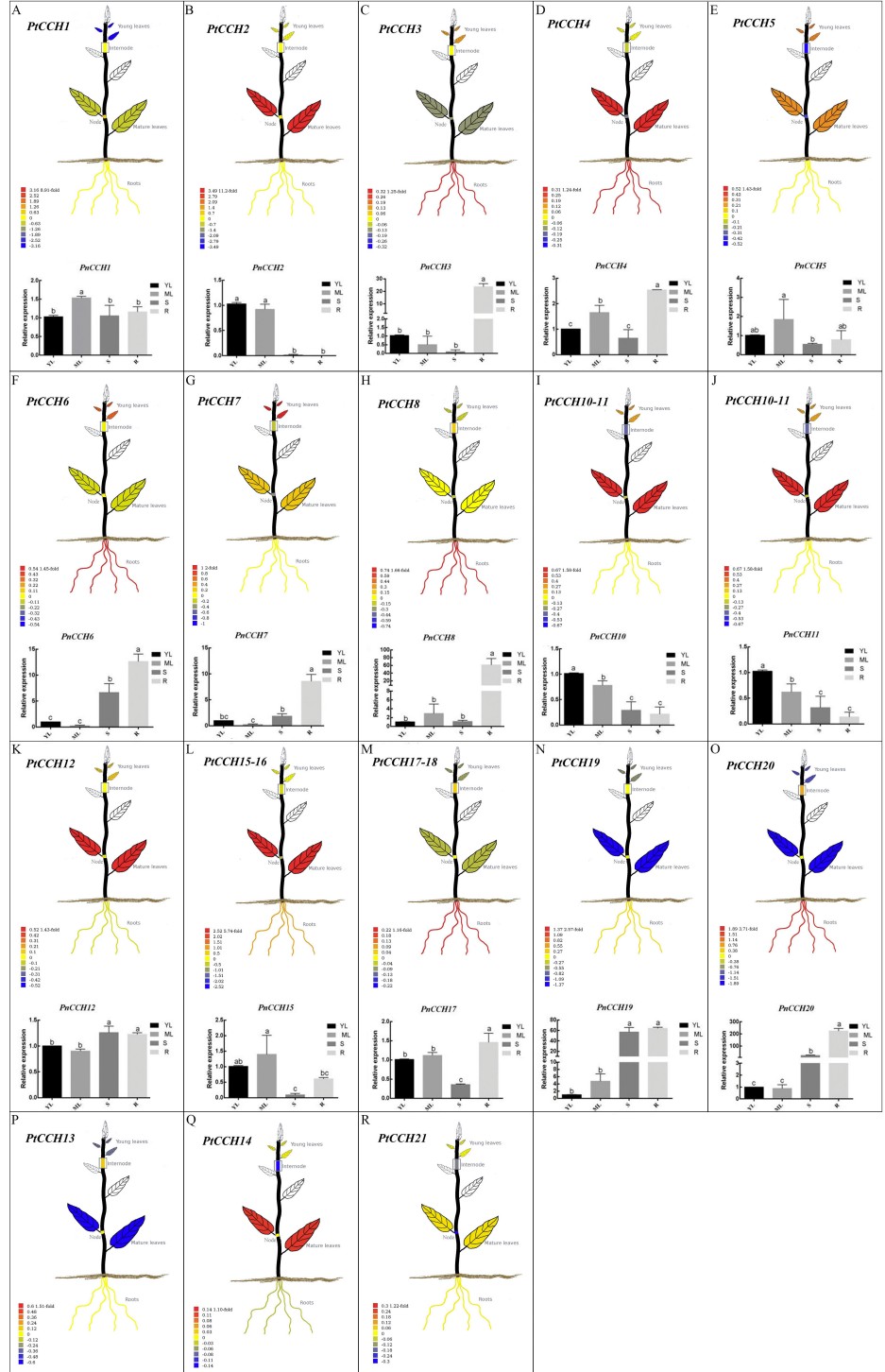

**Figure 5** **Tissue-specific expression profiles of *PtCCH* and *PnCCH* genes.** The visual images of *PtCCH* genes in *P. trichocarpa* were generated using the tissue-specific expression data of mature leaves, young leaves, roots, nodes, and internodes derived from http://PlantGenIE.org (*Sundell et al., 2015*). The bar graphs of *PnCCH* genes in *P. simonii* × *P. nigra* were generated by qRT-PCR. Expression values of young leaves (YL), mature leaves (ML), stems (S), and roots (R) were normalized to the relative levels of *PnUBQ7*. Data represent mean values of three replicates, error bars represent SE, and different letters represent statistical significant differences ($P < 0.05$) using duncan's test.

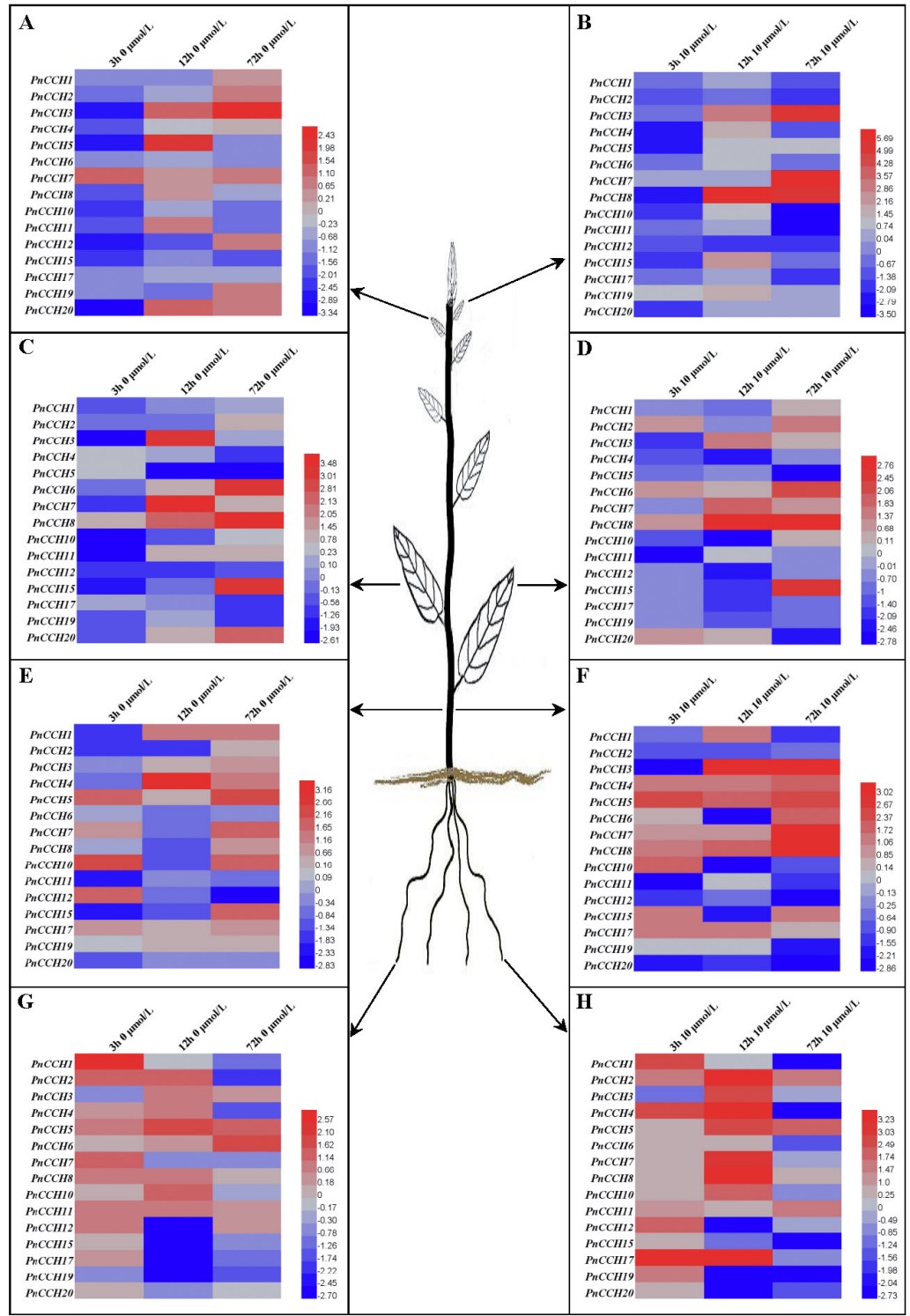

**Figure 6 Expression profiles of *PnCCH* genes in different tissues under copper stress.** (A), Expression profiles of *PnCCH* genes in young leaves under 0 μM CuSO₄ condition. (B), Expression profiles of *PnCCH* genes in young leaves under 10 μM CuSO₄ condition. (C), Expression profiles of *PnCCH* genes in mature leaves under 0 μM CuSO₄ condition. (D), Expression profiles of *PnCCH* genes in mature leaves under 10 μM CuSO₄ condition. (E), Expression profiles of *PnCCH* genes in (continued on next page...)

**Figure 6 (…continued)**
stems under 0 μM CuSO$_4$ condition. (F), Expression profiles of *PnCCH* genes in stems under 10 μM CuSO$_4$ condition. (G), Expression profiles of *PnCCH* genes in roots under 0 μM CuSO$_4$ condition. (H), Expression profiles of *PnCCH* genes in roots under 10 μM CuSO$_4$ condition. In the heatmaps, the genes are shown on the left, and the copper stress concentrations (0 μM CuSO$_4$ is deficiency copper condition, 10 μM CuSO$_4$ is excessive copper condition, and 0.5 μM CuSO$_4$ is control copper condition) and treatment times are indicated on the top. The black arrows indicate the tissues, young leaves, mature leaves, stems, and roots, which were tested. The $2^{-\Delta\Delta CT}$ method was used to analyze the expression levels of *PnCCHs* in different tissues under copper stress conditions. The values of log$_2$ (sample/control) upon to copper stress conditions and different treated times were calculated as the relative expression levels of every *PnCCH* genes. Scale bars are on the bottom right, and the different color of the cells in the heatmaps indicated the expression level of the treated samples went up or down compared with their controls.

*PnCCH19* had the highest expression levels in young leaves, roots/stems, respectively, while *PnCCH2* was most highly expressed in young and mature leaves (Figs. 5A–5R). The expression patterns of the remaining *PnCCHs* could not be measured by qRT-PCR.

## Expression patterns of *PnCCH* genes under copper stress

Previous studies showed that the function of AtCCH was copper-dependent, and that *AtCCH* mRNA was induced in the absence of copper and reduced with excess copper (*Himelblau et al., 1998*; *Shin, Lo & Yeh, 2012*). To investigate the functions of PnCCHs, the expression patterns of the *PnCCH* genes under copper stresses (deficiency and excessive) were analyzed by qRT-PCR. All the *PnCCH* genes exhibited expression variations in response to deficiency (0 μM CuSO$_4$) and/or excessive copper (10 μM CuSO$_4$). In different tissues, the expression levels of *PnCCH* genes relative to controls (0.5 μM CuSO$_4$) differed (Fig. 6, Table S5). In young leaves, the expression levels of *PnCCH3* were higher than the controls under either deficiency or excessive copper conditions for 72 h, and the differences were significant. *PnCCH7* and *PnCCH8* had increased expression levels with significant differences under excessive copper for 72 h and 12 h, respectively. The expression levels of *PnCCH2* and *PnCCH12* under excessive copper, and *PnCCH4*, *PnCCH6*, *PnCCH10*, *PnCCH15*, and *PnCCH17* under copper deficiency could not be induced (Figs. 6A–6B). In mature leaves, the expression levels of *PnCCH3* and *PnCCH7* increased significantly under copper deficiency for 12 h, and the expression levels of *PnCCH6*, *PnCCH8*, and *PnCCH15* were higher than the controls significantly under the deficiency or excessive copper conditions for 72 h. The expression levels of *PnCCH12* under copper deficiency and *PnCCH4*, *PnCCH5*, *PnCCH11*, *PnCCH12*, *PnCCH17*, and *PnCCH19* under the copper excessive condition were not induced (Figs. 6C–6D).

In stems, the expression levels of the *PnCCH* genes increased under copper deficiency, except for *PnCCH6*, *PnCCH11*, and *PnCCH20*; notably, the expression level of *PnCCH4* was distinctly higher than the control after treatment for 12 h. When treated with excess copper, *PnCCH3*, *PnCCH7*, and *PnCCH8* had increased transcript levels significantly in stems compared with those of the controls after treatment for 72 h, while the expression levels of *PnCCH2*, *PnCCH12*, *PnCCH19*, and *PnCCH20* were not be induced by excessive copper treatment. Unlike its lack of response under copper deficiency, the expression of *PnCCH6* was higher than the controls significantly after excessive copper treated for 72 h (Figs. 6E–6F). In roots, the expression of most *PnCCH* genes was induced under

the copper deficiency condition after treatment for different times, except for *PnCCH19*, which had lower expression levels relative to the controls significantly. The expression levels of *PnCCH12*, *PnCCH15*, *PnCCH17*, and *PnCCH19* were significantly decreased compared with controls after copper deficiency treatment for 12 h, and the expression levels of *PnCCH1*, *PnCCH4*, *PnCCH17* and *PnCCH19* were decreased significantly after copper deficiency treatment for 72 h (Fig. 6G). Under excessive copper treatment, the expression levels of *PnCCH1*, *PnCCH4*, *PnCCH6*, *PnCCH15*, *PnCCH17*, *PnCCH19*, and *PnCCH20* were lower than those of the controls significantly after treated for 72 h; and the expression levels of *PnCCH4* and *PnCCH17* increased distinctly after treatment for 3 h and 12 h (Fig. 6H).

## DISCUSSION

In this study, we identified 21 *PtCCH* genes in the *P. trichocarpa* genome. The chromosomal location and duplication analysis indicated that the *PtCCH* gene family may have arisen as the result of the whole-genome duplication and tandem duplication event, according to a genome-wide analysis of other gene families in *P. trichocarpa* (*Liu et al., 2012*; *Lan, Gao & Zeng, 2013*). Five pairs of *PtCCH* genes were in tandem repeat locations on the chromosomes, while *PtCCH3* and *PtCCH8/9*, as well as *PtCCH5* and *PtCCH13*, were located in a pair of paralogous blocks and could be regarded as the direct result of the salicoid duplication event. Nine *PtCCH* genes were located in WGD duplicate blocks but lacked the corresponding duplicate genes (Fig. 2), which suggested that the loss event might have happened after the whole-genome duplication, or these nine genes might have arisen after the salicoid duplication event (*Lan, Gao & Zeng, 2013*). In addition to the similarity of their amino acid sequences (89%), PtCCH1 and PtCCH19 were on the same branch of the phylogenetic tree and had highly conserved motifs, but *PtCCH1* and *PtCCH19* genes were located outside of the WGD duplicated blocks. It was deduced that these two genes might be the products of segment duplication. All these data indicated that both WGD and tandem duplication events may have played important roles in the expansion of *CCH* gene family in the *Populus* genome. To date, only one *CCH* gene has been reported in *A. thaliana* (ATU88711), *O. sativa* (AF198626), and *Glycine max* (AF198627), and the function of AtCCH is more clearly defined than the other two CCHs (*Himelblau et al., 1998*; *Shin, Lo & Yeh, 2012*). Large gene families formed by natural selection may contain more functional members associated with speciation or adaptation (*Lynch & Conery, 2000*; *Demuth & Hahn, 2009*). The expansion of the *CCH* family in the *Populus* genome might reflect the special roles played by these genes related to the powerful adaptability against physical and biotic stresses of this perennial woody species, as has been suggested for other gene families in *Populus* (*Hu et al., 2012*; *Liu & Widmer, 2014*).

Recently, some *CCH* genes with the conserved N-terminal metal binding motif MXCXXC and lysine-rich region have been identified in plants; for example, *PoCCH* (AY603358) in *Populus* (*Populus alba* × *P. tremula* var. *glandulosa*), *HbCCH1* (GU550955) in *Hevea brasiliensis*, and LeCCH (AY253832) in *Lycopersicon esculentum*. However, all of these genes lacked the C-terminal extension that existed in previously described CCH type

copper chaperones in plants that seemed to be related to copper intercellular transport from the senescent tissues (*Himelblau et al., 1998*; *Company & González-Bosch, 2003*; *Lee et al., 2005*; *Li et al., 2011*). In our study, all the deduced PtCCH proteins contained the MXCXXC motifs and the C-terminal extended sequences. The MXCXXC motifs were located in the first loop of the βαββαβ structural fold of PtCCHs. The N-domain of ATX1-like copper chaperones in plants, such as AtCCH and AtATX1, contributed to the copper chaperone function and antioxidant protection against oxidative damage, as described in the yeast Atx1 and human HAH1 counterparts (*Himelblau et al., 1998*; *Shin, Lo & Yeh, 2012*). The antioxidant and copper chaperone properties of PtCCHs could be investigated by identifying whether they are capable of complement the growth defects of yeast *atx1Δ* mutant cells on iron deficient medium, as well as the aerobic lysine growth deficiency of yeast *sod1Δ* mutants. Furthermore, 10 of the 21 PtCCHs identified in this study had the lysine-rich region, which was proposed to be involved in recognition and interaction with its target proteins. It was demonstrated that the lysine-rich region of yeast ScAtx1 might be associated with protein–protein interactions mediated by the electrostatic attraction between the lysine residue of Atx1 and acidic residues of Ccc2, a copper transporting ATPase (*Lamb et al., 1999*; *Mira et al., 2001b*). The results of yeast two-hybrid experiments indicated that AtATX1 could interact with the heavy metal P-type ATPases HMA5 and RAN1 of *Arabidopsis*, as well as AtCCH after deleting the C-terminus (*Andrés-Colás et al., 2006*; *Shin, Lo & Yeh, 2012*). Further experiments are required to verify whether PtCCHs have the important role of delivering copper to P-type ATPase, and to clarify the specific functions of their plant-exclusive C-domains.

In our study, some *PtCCH* genes that had a near relationship in the evolutionary tree had the same exon/intron organization, and the distributions and arrangements of the putative motifs in the deduced proteins were similar. In addition, the unique motifs and their arrangements in different subgroups might be correlated with the specific functions of the PtCCH family members. Except for motif 3, which was present in all the PtCCHs and contains the MXCXXC site that is essential for copper binding, all the other motifs of PtCCHs had no related report in copper binding and transporting. So we speculated that most motifs might just conserved sequences between CCH proteins, and the functions of them need further identification. We could not use the domain combination analysis to identify novel functions of PtCCH proteins yet. According to the up-to-date genome annotations, we found 21, 13, and 12 putative *CCH* genes in *P. trichocarpa*, *A. thaliana*, and *O. sativa*, respectively. All the CCHs from these three species were classified into three groups in the phylogenetic tree (Fig. 4). The expansion of the *CCH* gene family in poplar, compared with *Arabidopsis* and rice, may be related to the stronger growth and adaptability of woody plants than herbaceous plants on the land, and also reflected the specialized roles played by these genes in copper transport and copper homeostasis regulation (*Tuskan et al., 2006*; *Zhang et al., 2015*). In addition, the *CCH* genes of *Populus* and *Arabidopsis* had fewer than three introns. According to some studies, genes which can rapidly change their expression levels in response to stresses generally contain fewer introns to minimize the cost of transcription and other molecular processes (such as splicing), and highly expressed
genes appear to have developed short introns during the natural selection (*Castillo-Davis et al., 2002*; *Jeffares, Penkett & Bähler, 2008*).

Furthermore, the reservation of duplicated genes may be due to either genetic redundancy or function renewal (*Lynch & Conery, 2000*; *Dean et al., 2008*). Functional novelty includes neo-functionalization by obtaining new functions, or sub-functionalization by dividing the functional modules, and the functional complementation of both duplicates could perform the original roles of their ancestral gene (*Liu & Widmer, 2014*). To evaluate the retention modes of *PtCCH* genes after duplication, we constructed visual images using the expression data of *P. trichocarpa* in http://PlantGenIE.org (Figs. 5A–5R). The results showed that these *PtCCH* genes in *Populus* displayed expression divergence in mature leaves, young leaves, roots, nodes, and internodes. For instance, *PtCCH3* and PtCCH8/9, as well as *PtCCH5* and *PtCCH13*, was deduced be the result of whole-genome duplication event. *PtCCH3* and *PtCCH8*, *PtCCH13*, *PtCCH15* had higher expression levels in roots, internodes, mature leaves, respectively. The redundancy or renewal functions of *PtCCH3* and PtCCH8, as well as *PtCCH5* and *PtCCH13*, need further identification. In *Arabidopsis*, *AtCCH* mRNA was expressed in roots, stems, leaves, inflorescences, and siliques, and the CCH protein was located mainly along the vascular bundles of senescent leaves, petioles, and stem sieve elements (*Himelblau et al., 1998*; *Mira, Martínez-García & Peñarrubia, 2001a*). Whether all the *PtCCH* genes underwent neo-functionalization or sub-functionalization remains to be explored.

The comparative analysis of the tissue-specific expression profiles of the *PnCCHs* detected by qRT-PCR and the *PtCCHs* from http://PlantGenIE.org showed that consistent expression patterns between *PtCCHs* and *PnCCHs* existed mostly in roots, while the biggest difference in expression was in the young leaves. All members of subgroup SIII in the phylogenetic tree of *PtCCHs* (Fig. 3A), were highly expressed in mature leaves (*Sundell et al., 2015*), and the *PnCCHs* (*PnCCH2*, *PnCCH10*, and *PnCCH11*) that corresponded to most of them (*PtCCH2*, *PtCCH10*, and *PtCCH11*) had high expression levels in young leaves. Five *PtCCHs* (*PtCCH1*, *4*, *17*, *18* and *PtCCH19*) that clustered in subgroup SII, as well as four *PnCCHs* (*PnCCH4*, *12*, *17*, and *PnCCH19*), had high expression levels in roots (Figs. 5D, 5K, 5M and 5N). *P. trichocarpa* is cultivated mainly in western North America, and *P. simonii × P. nigra* is a hybrid poplar widely planted in northern China. The discrepancy and coherence of expression patterns between *PtCCHs* and *PnCCHs* might be caused by species-specificity between *P. trichocarpa* and *P. simonii × P. nigra*. Further investigations are required to clarify the roles of these putative copper chaperones in metal transport.

Expression patterns are usually associated with the functions of genes (*Guo et al., 2008*; *Ma et al., 2016*). The analysis of differential expression profiles of *PnCCH* genes could provide important information about their function specializations. In this study, we analyzed the expression profiles of *PnCCHs* in response to copper stress conditions. In different tissues, *PnCCH* genes exhibited variations in expression under deficiency or/and excessive copper conditions relative to controls (Fig. 6, Table S5). For example, tissue-specific expression profiles indicated that *PnCCH3*, *PnCCH4*, *PnCCH6*, *PnCCH7*, *PnCCH8*, *PnCCH12*, *PnCCH17*, *PnCCH19*, and *PnCCH20* were highly expressed in roots. The expression levels of *PnCCH3*, *PnCCH4*, *PnCCH7*, *PnCCH8*, and *PnCCH17* increased

distinctly after being treated with excessive copper for 12 h, while the expression levels of *PnCCH3*, *PnCCH6*, and *PnCCH7* increased only slightly under copper deficiency condition treated for 12 h. In addition, the expression levels of *PnCCH19* and *PnCCH20* decreased under both deficiency and excessive copper conditions after being treated for more than 12 h. The expression level of *PnCCH6* increased gradually under copper deficiency condition with increasing treating times, while it reduced significantly under excessive copper after treated for 72 h. For *PnCCH17*, the expression level decreased under copper deficiency condition with increasing treating times, while it first increased and then decreased significantly under the excessive copper condition (Figs. 6G–6H). These results indicated that PnCCHs might take part in a complex copper homeostasis network of different tissues. To date, only one *CCH* gene has been identified in *Arabidopsis*, and although the function of the encoded copper chaperon was studied to some extent (*Himelblau et al., 1998*; *Mira et al., 2004*; *Shin, Lo & Yeh, 2012*), the roles of AtCCH and its CTD need to be explored further. Because of the stronger adaptability of trees on land, we deduced that the CCH proteins identified in *Populus* might have more complex functions in copper cellular homeostasis than AtCCH. The mechanisms of the actions of PnCCHs in copper binding and transport processes require further exploration combining the tissue-specific expression profiles.

Until now, only a few *CCH* genes were identified in plants, and none in *Populus*. This study has laid a foundation for our further identification the functions of *CCH* gene family members in poplar, as well as for clarification the relationship between structure and function of *CCH* genes in *Populus*. Our research might also provide some references for understanding the complex mechanisms of poplar CCH proteins in copper homeostasis.

## CONCLUSION

Bioinformatics technology was used to screen and analyze *CCH* gene family in the woody plant, *P. trichocarpa*. All 21 PtCCHs had the metal-binding motif MXCXXC, βαββαβ secondary structure, and the extended C-terminal. The *PtCCH* genes that had a near relationship in phylogenetic tree generally had similar exon/intron structures and the conserved motifs in the encoded proteins. The tissue-specific expression traits of *PtCCHs* in *P. trichocarpa* and *PnCCHs* in *P. simonii* × *P. nigra* had discrepancy and coherence. In different tissues, the expression profiles of *PnCCH* genes under deficiency or/and excessive copper conditions exhibited variations. The present study opens up a new perspective to study copper homeostasis in poplar, and may also provide useful genetic resources for further research on heavy metal phytoremediation.

### Funding

This study was supported by the National Natural Science Foundation of China (31470664 and 31370662), and the Fundamental Research Funds for the Central Universities

(2572017EA05). The funders had no role in study design, data collection and analysis, decision to publish, or preparation of the manuscript.

### Grant Disclosures

The following grant information was disclosed by the authors:
National Natural Science Foundation of China: 31470664, 31370662.
Fundamental Research Funds for the Central Universities: 2572017EA05.

### Competing Interests

The authors declare there are no completing interests.

### Author Contributions

- Zhiru Xu conceived and designed the experiments, contributed reagents and materials, wrote the paper, prepared some figures and/or tables, reviewed drafts of the paper.
- Liying Gao and Mengquan Tang performed the experiments.
- Chunpu Qu, Jiahuan Huang and Qi Wang analyzed the bioinformatics and qRT-PCR data.
- Chuanping Yang provided valuable suggestions and comments.
- Guanjun Liu conceived and designed the experiments, contributed reagents/materials/analysis tools, reviewed drafts of the paper.
- Chengjun Yang performed most of the bioinformatics analysis.

### DNA Deposition

The following information was supplied regarding the deposition of DNA sequences:
The sequences have been uploaded as a Supplemental File.

### Data Availability

The raw data has been supplied as a Supplemental File.

### Supplemental Information

Supplemental information for this article can be found online at http://dx.doi.org/10.7717/peerj.3962#supplemental-information.

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
