# Peer review of "Genome-wide identification and expression profile analysis of CCH gene family in Populus"

_PeerJ, doi:10.7717/peerj.3962_

## Round 0.1 · original submission · Minor Revisions

Both reviewers are positive about the research presented in your manuscript, but have some very useful suggestions to improve the presentation of your results. For example, including more information about statistical analyses is necessary for readers to assess the statistical significance of your results. Please address all of the comments/suggestions of both reviewers by either making the suggested changes or providing a rationale for not making specific suggested changes.

Reviewer 1 ·

Basic reporting

Figure 6 (RT-qPCR results) should include the raw data or bar plots and the statistical significance of the difference should be discusses. It is not easy to identify the level of difference from heatmaps. In addition, because untreated and treated samples are in separate plots, they cannot be directly compared due to separate normalization.

Experimental design

The article will fall within the scope of PeerJ. The analyses are routinely used and are well done. There are some issues however. See specific comments related to experimental design are below.

1. The definition or identity of CCH gene family is not clear and objective. Is the MXCXXC motif and the secondary structure pattern not present in any other gene family? Is there a minimum sequence identity that authors looked for. A single criteria of BLASTp with an E-value seems arbitrary. It should be clearly discussed what makes a CCH family unique, and use that criteria.

2. Is the induction of expression level differences upon Cu stress statistically significant? It is not clear from the heatmaps. If it is not significant, it should be clearly stated and discussed.

3. It is not clear how the A. thaliana, and O. sativa CCH genes were identified. Will a similar sequence search conducted for CCH in these genomes. The criteria should ideally be same for all genomes.

4. It is known that a single query may not be enough to cover all the genes in a gene family and the choice of query also affects the family coverage (see e.g. Kaushik et al. PLOS ONE 2013). It can be tested by a sequence search using all A. thaliana CCH if any new CCH proteins are identified in any of the 3 organisms.

5. Are the functions of the predicted motifs known? And based on the motifs, can a new functions of CCH in Populus be identified? Domain combination analysis can also be used to identify novel functions of CCH if any.

6. Has the lineage leading to the evolution of Populus experienced multiple Whole Genome Duplications? How does genomic distribution of CCH relate to other WGD events in the past?

7. Is there any specific property in ohnolog PtCCH and the ones that are retained from SSDs? It will be interesting to discuss.

8. Has the expression pattern diverged between the leaves in Pt and PnCCH? What could be the reasons and implications for such a divergence.

Validity of the findings

The finding would appeal to limited audience interested in CCH family in plants. Most of the article is very descriptive. If the results are put in a broader context, it will make the manuscript more appealing to broad audience.

Reviewer 2 ·

Basic reporting

The manuscript by Xu et al. describes the identification of genes encoding copper chaperone (CCH) family proteins in Populus. Based on the known P. trichocarpa genomic sequence, they perform a bioinformatic analysis and identify 21 proteins with sequence and structural similarity to Arabidopsis CCH, analyze the chromosomal location of the corresponding genes, and perform a phylogenetic analysis including proteins from rice and Arabidopsis. They also analyze the expression of several of the Populus genes in different organs under different conditions (control, copper deficiency, copper excess). Their main conclusion is that the CCH family is expanded in Populus and they suggest that this may be related to the ability to grow on contaminated areas and accumulate metals.

The manuscript is correctly written, with references to the relevant literature. The bioinfomatic and experimental analysis was properly performed, although some corrections and clarifications are required, as described below.

Figures and figure legends:
-Fig. 2 legend: Please, explain the meaning of the different boxes connecting the chromosomes.
-Fig. 3C: References to motifs are too small and difficult to read. Please, refer in the legend that the sequences of the motifs are listed in Table S4. Also, are these motifs similar to any known motif, thus giving clues to their possible functions, or are they just conserved sequences between CCH proteins?
-Fig. 3C legend: explain the meaning of S1 to S3.
-Please, add bootstrap values to the trees, or an indication of how significant the different branches are. Related to this, do the trees shown in Figs. 3 and 4 have the same structure? If not, how do the authors explain this? How do clades S1 to S3 in Fig. 3 relate to clades SI-SIII observed in Fig. 4? Ideally, the labeling of clades should be the same (S1-S3 or SI-SIII).
-Fig. 4: I guess there is a mistake in the labeling of some groups (blue and green in the upper left part should be SIII instead of SII).
-Fig. 5: The labels in the bar graphs are difficult to see (too small). The legend should be more descriptive, so that the reader can understand the figure without referring to the text. This applies to all figures. Please revise this.
-Fig. 6: Relative expression is shown. Relative to what? Untreated controls? Please, clarify this. Also, give a hint of the statistical significance of the results (for example, include an asterisk in those rectangles where changes in expression are statistically significant).

Discussion:
Lines 425-434, explaining primer design and why the expression of some genes was not measured, should be removed from here and included under Results.

Conclusions:
The Conclusions are not properly written. English is poor and there is repetition of concepts already mentioned in other sections. Re-write or, preferably, remove this section, since it does not add much to the paper.

Experimental design

I find that a basic question that is not correctly addressed in the manuscript is that of the function of members of the CCH family. The authors refer ambiguously to the CCH gene from Arabidopsis, which has been functionally characterized, and then identify 13 CCH genes in this species. Are there any studies on these other CCH genes? Which one of these 13 is the “real” CCH? (I guess it is AtCCH6). These matters should be discussed in the manuscript. Also, I suggest that the authors perform complementation studies in yeast to analyze if representative Populus CCH proteins from different clades are functionally equivalent to Arabidopsis CCH. This would add a functional significance to the expansion of the family.

Validity of the findings

No comment

---

## Round 0.2 · Minor Revisions

General comment:

Thank you for your detailed response to both reviewers’ comments in your rebuttal letter. In some cases, you answer a reviewer’s comments very completely in your rebuttal letter, but do not include all the same information in your revised manuscript. Please go over your responses to the reviewers’ questions and comments in your rebuttal letter and make sure that the information presented in the rebuttal letter is also in the revised manuscript. In this way readers of your manuscript who have the same questions as the reviewers will easily be able to see the answers to those questions.

Specific comments:

Table S5: Thank you for including a new table presenting more data regarding gene expression levels. To ensure that readers are able to understand this table, please describe how the numbers presented in the table legend were calculated and whether the +/- intervals indicate standard deviations or standard errors. Also, at the bottom of the table, please state which values are being compared with which other values in calculating p values and what type of test was conducted to determine p values.

Figures 5 and 6: The font is small in most of the figures, but is particularly small in figures 5 and 6. Especially in figure 5, it is impossible to read what is written without greatly magnifying the figure. Please try to make the font larger in these figures.

---

## Round 0.3 · accepted · Accept

Thanks for submitting a revised version of your manuscript, and for trying to increase the font size in your figures. I agree that this can be difficult to do with large figures containing multiple panels.